# An Efficient and Quick Analytical Method for the Quantification of an Algal Alkaloid Caulerpin Showed In-Vitro Anticancer Activity against Colorectal Cancer

**DOI:** 10.3390/md20120757

**Published:** 2022-11-30

**Authors:** Nazli Mert-Ozupek, Gizem Calibasi-Kocal, Nur Olgun, Yasemin Basbinar, Levent Cavas, Hulya Ellidokuz

**Affiliations:** 1Department of Basic Oncology, Institute of Health Sciences, Dokuz Eylül University, Izmir 35340, Turkey; 2Department of Translational Oncology, Institute of Oncology, Dokuz Eylül University, Izmir 35340, Turkey; 3Department of Pediatric Oncology, Institute of Oncology, Dokuz Eylül University, Izmir 35340, Turkey; 4Department of Chemistry, Faculty of Sciences, Dokuz Eylül University, Izmir 35390, Turkey; 5Department of Preventive Oncology, Institute of Oncology, Dokuz Eylül University, Izmir 35340, Turkey

**Keywords:** bioinvasion, *Caulerpa*, colorectal cancer, HPTLC, MALDI-TOF/MS

## Abstract

Biological invasion is the successful spread and establishment of a species in a novel environment that adversely affects the biodiversity, ecology, and economy. Both invasive and non-invasive species of the *Caulerpa* genus secrete more than thirty different secondary metabolites. Caulerpin is one of the most common secondary metabolites in genus *Caulerpa*. In this study, caulerpin found in invasive *Caulerpa cylindracea* and non-invasive *Caulerpa lentillifera* extracts were analyzed, quantified, and compared using high-performance thin layer chromatography (HPTLC) for the first time. The anticancer activities of caulerpin against HCT-116 and HT-29 colorectal cancer (CRC) cell lines were also tested. Caulerpin levels were found higher in the invasive form (108.83 ± 5.07 µg mL^−1^ and 96.49 ± 4.54 µg mL^−1^). Furthermore, caulerpin isolated from invasive *Caulerpa* decreased cell viability in a concentration-dependent manner (IC50 values were found between 119 and 179 µM), inhibited invasion-migration, and induced apoptosis in CRC cells. In comparison, no cytotoxic effects on the normal cell lines (HDF and NIH-3T3) were observed. In conclusion, HPTLC is a quick and novel method to investigate the caulerpin levels found in *Caulerpa* extracts, and this paper proposes an alternative utilization method for invasive *C. cylindracea* due to significant caulerpin content compared to non-invasive *C. lentillifera*.

## 1. Introduction

Biological invasion (or bioinvasion) is a biological phenomenon defined as the spread and establishment of a species in a novel environment successfully [1,2]. Bioinvasion is often driven by human activities (such as aquaculture, shipping, transportation, and ongoing climate change) [1,3,4] and it causes ecological, economic, health, and social damages by causing deterioration of biodiversity and ecosystem functioning, the spread of diseases, and decrease in the quality of the economy [1,2]. Macroalgae, or seaweeds, offer a considerable source of bioactive chemicals (especially secondary metabolites) with potential uses in such industries as cosmetics, food, pharmacy, and medicine. The *Caulerpa* genus, the green siphonous macroalga, belongs to the Caulerpaceae family with 97 species [5]. Among the secondary metabolites in *Caulerpa* spp., the bisindole alkaloid caulerpin has antinociceptive and anti-inflammatory [6], antimicrobial activities against *Escherichia coli, Staphylococcus aureus,* and *Streptococcus* sp. [7]; inhibition of monoamine oxidase-B [8], antituberculosis activity [9]; and antinociception [10] activities. Furthermore, it was stated by in silico studies that it has antiviral activity against bovine viral diarrhea virus, herpes simplex virus, chikungunya virus, and SARS-CoV-2 [11,12,13,14,15,16,17]. It was reported that caulerpin has a hepatocellular carcinoma peroxisome proliferator activating receptor alpha and peroxisome proliferator activating receptor gamma activity in vivo and in vitro [18]. The role of secondary metabolites found in *Caulerpa* spp. in cancer metabolism is currently being studied, as well as the complex modulation network induced in AMPK modulation and endoplasmic reticulum stress, inhibition of hypoxia-inducible factor 1, protein tyrosine phosphatase1B inhibition and metabolic reprogramming, apoptosis and cell cycle arrest in cancer cells [5].

The method for the determination of caulerpin level is very limited in the literature. To determine the caulerpin found in tissues and extracts quantitatively, high-performance liquid chromatography (HPLC) or ultra-performance liquid chromatography coupled with mass spectrometry (UPLC/MS) are commonly used analytical methods [19,20,21]. However, thin-layer chromatography (TLC) has commonly been used to detect the caulerpin in extracts qualitatively. As post-treatments such as ceric sulfate spraying are required in analyzing caulerpin by TLC [12], it is hard to determine the caulerpin level quantitatively by this method (TLC). Diversely, high-performance thin-layer chromatography (HPTLC) provides more information about the compound quantitatively and qualitatively [22]. Analyzing the natural compounds by HPTLC has several advantages, such as high-throughput screening, low-volume sample preparation, cost-effective analysis, high sensitivity, short analysis and development time, and good reproducibility [22]. Thus, HPTLC is of great importance for the determination of compounds using planar chromatographic methods.

Colorectal cancer (CRC) is the third most common cancer type and the second cause of cancer-related death worldwide. In 2020, the annual incidence of CRC was reported as more than 1.9 million [23,24]. The development of CRC depends on multiple factors, such as lifestyle (dietary patterns, obesity, smoking, alcohol, etc.), family history and genetics (inflammatory bowel disease, colon polyps, diabetes mellitus, etc.), and personal factors (age, gender, gut microbiota, socioeconomic factors, etc.) [25]. Traditional CRC treatment involves chemotherapy and surgery. Although chemotherapeutics are commonly used for the treatment of CRC, side effects of chemotherapeutics that relate to non-specific actions (not only the tumor cells but also non-malignant cells) are the limit of the therapy [26]. Thus, using natural compounds as CRC treatment agents are of great importance.

The study aimed to determine the caulerpin level in different extracts of invasive *Caulerpa cylindracea* and non-invasive *Caulerpa lentillifera* by using HPTLC quantitatively, validate the HPTLC method by using the International Council for Harmonization (ICH) Q2 (R1) recommendations, characterize the isolated samples by using MALDI-TOF/MS, and investigate the biological activities of caulerpin against CRC cell lines using in vitro methods.

## 2. Results

### 2.1. HPTLC Validation of the Method for Quantification of Caulerpin

The different extracts, *Caulerpa* spp. (CC48, CC72, CL48, CL72, CCM, CLM) were analyzed using HPTLC. Rf value was defined as 0.41. According to the results, the highest and lowest caulerpin concentrations were found in CC48 and CC72, respectively. The caulerpin levels of CC48, CC72, CL72, CCM, CLM were found at 453.46 ± 21.10 ng µL^−1^ (108.83 ± 5.07 µg g^−1^), 402.04 ± 18.93 ng µL^−1^ (96.49 ± 4.54 µg g^−1^), 35.16 ± 16.07 ng µL^−1^ (8.44 ± 3.86 µg g^−1^), 562.14 ± 26.42 ng µL^−1^ (112.43 ± 5.29 µg g^−1^), and 42.09 ± 8.20 ng µL^−1^ (8.42 ± 1.64 µg g^−1^), respectively (Figure 1 and Figure 2). However, caulerpin was not found in the CL48 extract. All data are given in Table 1. The LOD and LOQ values for caulerpin were found as 20.47 and 67.56 ng µL^−1^, respectively. Accuracy (recovery percentage; R%) and precision were found as 89.81% and 19.2%, respectively. Related data are given in Table 2.

### 2.2. MALDI-TOF/MS Result of Caulerpin

Caulerpin was analyzed using MALDI-TOF/MS (Figure 3). In Figure 3, the most abundant peak was observed at 398.11 m/z, and this molecule was defined as caulerpin, and at 871.52 m/z could be chlorophyll a without the magnesium ion [27].

Since the 871 m/z fragment was observed in the MALDI-TOF/MS result of the isolated caulerpin (Figure 3), an HPLC analysis was performed for the isolated caulerpin by using the method of Turhan and Cavas [19], and the caulerpin peak at 30.9 min was observed. Furthermore, there were no other peaks during the HPLC analysis (Figure 4).

### 2.3. The Effect of Caulerpin on Colorectal Cancer Cell Line Viability

The cytotoxic activity of caulerpin against HCT-116 and HT-29 cell lines treated with various concentrations of caulerpin was measured by using WST-1 reagent. After 48-h incubation, IC_50_ concentrations were calculated as 119 and 179 µM for HCT-116 and HT-29 cell lines, respectively. Furthermore, after 48-h incubation, IC_10_ concentrations were 73 µM for HCT-116 and 66 µM for HT-29 cell lines (Figure 5A,B). For the NIH/3T3 cell line, the IC10 doses of the caulerpin did not significantly reduce the cell viability at the 24th hour compared to the control group. However, for the NIH/3T3 cell line, the IC_10_ dose of the HCT-116 cell line (* *p* = 0.04) and the IC_50_ dose (* *p* = 0.04) dose of the HCT-116 cell line showed significantly reduced viability at 48 h compared to the control. On the other hand, the IC_50_ doses did not reach the half maximal viability at the 48th hour (HCT-116 IC_50_ dose cell viability: 67.5% and HT-29 IC_50_ dose cell viability: 68.8%) (Figure 5C).

For the HDF cell line, the IC_50_ dose of the HT-29 cell line (*p* = 0.04) and the IC_50_ dose of the HCT-116 cell line (* *p* = 0.04) significantly reduced the cell viability at the 24th hour compared to the control group. Furthermore, the IC_10_ doses of the caulerpin did not significantly reduce the cell viability at the 24th hour compared to the control group. On the other hand, the IC_50_ doses did not reach the half maximal viability at the 48th hour (HCT-116 IC_50_ dose cell viability: 84.7% and HT-29 IC_50_ dose cell viability: 76.6%) (Figure 5D).

The long-term effect of caulerpin on the proliferative capacity of HCT-116 and HT-29 cell lines was evaluated by a colony formation assay after 15 days of treatment with IC_10_ and IC_50_ doses of caulerpin. The results revealed that after incubation with caulerpin, the colony formation of HCT-116 and HT-29 cell lines was inhibited (**, *p* = 0.0043 for HCT-116 cell line and **, *p* = 0.0010 for HT-29 cell line (Figure 5E,F).

### 2.4. Inhibition of Migration and Invasion and Inducing Apoptosis of HCT-116 and HT-29 Cell Lines by Caulerpin

Migration is the main step of metastasis, thus, a wound healing assay was performed to examine the effect of caulerpin on the migration of the HCT-116 and HT-29 cell lines. After 48-h incubation, the control group exhibited 53.5% closure for the HCT-116 cell line. IC_10_ and IC_50_ doses of caulerpin showed 19.5% and 23.6% closure for the HCT-116 cell line, respectively (for the 24th hour, * *p* = 0.0319 and ** *p* = 0.0080; for the 48th hour, * *p* = 0.0407 and ** *p* = 0.0017). On the other hand, after 48-h incubation, the control group, IC_10_, and IC_50_ doses of caulerpin exhibited 41.8%, 5.78%, and 13.9% closure for HT-29 cell line, respectively (for the 24th hour, *** *p* = 0.0001). The results were given in Figure 6A,B.

The invasion activity of caulerpin on HCT-116 and HT-29 cell lines was investigated at the 6th, 24th, and 48th hours (Figure 6C,D). As shown in Figure 6C, caulerpin inhibited invasion on the HCT-116 cell line at the 48th hour (for the 6th hour, * *p* = 0.0436; for the 24th hour, ** *p* = 0.0072). The xCELLigence RTCA DP result reveals that the IC_50_ dose of caulerpin is quite effective on HT-29 cell line invasion, as shown in Figure 6D (for 6th hour, ** *p* = 0.0084 and **** *p* < 0.0001; for 48th hour, ** *p* = 0.0048).

After incubation with the caulerpin for 48 h, the Hoechst/PI staining protocol was applied to investigate the apoptotic activity. The results show that after 48 h exposed to IC_10_ and IC_50_ doses of caulerpin, the percent of apoptotic cells increased to 11.42 ± 2.67 and 34.28 ± 1.65 for HCT-116 cells, respectively (* *p* = 0.0102). Furthermore, in the treatment of IC_10_ and IC_50_ doses of caulerpin, the percent of apoptotic cells was increased to 11.42 ± 2.67 and 34.28 ± 1.65 for HCT-116 cells, respectively, and 10.22 ± 2.67 and 41.90 ± 1.65 for HT-29 cells, respectively (** *p* = 0.0047) (Figure 7).

## 3. Discussion

In this paper, caulerpin found in invasive *C. cylindracea* and non-invasive *C. lentillifera* extracts was analyzed and quantified using HPTLC for the first time. Among the two different extractions (soxhlet and maceration) procedures prepared from two different species, the caulerpin levels were compared. The anticancer activity of caulerpin against colorectal cancer cell lines was also tested. The results reveal that caulerpin levels were higher in invasive species. Furthermore, caulerpin was found to be an effective compound against CRC. HPTLC is a fast and novel method to investigate the caulerpin level from *Caulerpa* spp. (especially invasive *C. cylindracea*). Utilizing this bisindole alkaloid-based secondary metabolite against CRC treatment is promising.

Extraction is the first and most important step in the isolation of natural compounds [26,27]. Soxhlet extraction is defined as hot and continuous extraction that allows for obtaining more chemicals from the material (algae, plants, animals, etc.) [28,29]; on the other hand, maceration is a traditional method that is an easy and short procedure. In this study, the highest and the lowest caulerpin levels were found as 562.14 ± 26.42 ng µL^−1^ (112.43 ± 5.29 µg g^−1^) and 402.04 ± 18.93 ng µL^−1^ (96.49 ± 4.54 µg g^−1^) in CCM and CC72 extracts, respectively. The caulerpin levels were higher in *C. cylindracea* extracts than in *C. lentillifera* extracts (*p* < 0.05). In the literature, caulerpin levels were determined in different organisms by using HPLC [18,20,21,29]. According to the study by Terlizzi et al., the caulerpin was isolated from *C. racemosa* and the level was found as 50.4 µg g^−1^ fresh weight [30]. Mao et al. found the caulerpin levels in *C. taxifolia* to be 528.7 µg g^−1^ dry weight (0.05% dry weight) [31]. Lucena et al. isolated caulerpin from *C. racemosa* with a yield of 0.1% dry weight. They obtained 5 g caulerpin from 5 kg-dried samples [32]. In the study of Turhan and Cavas, caulerpin levels in *C. cylindracea* were found between 1090 ± 0.01 and 1820 ± 0.21 µg g^−1^ dry weight [21]. They also found that the season affected the caulerpin levels among *C. cylindracea* samples. Nagappan and Vairappan (2014) found that the caulerpin levels in *C. racemosa* var. *clavifera* f. *macrophysa*, *C. racemosa* var. *laetevirens,* and *C. lentillifera* were 2.6%, 1.8%, and 3.5% mg, respectively [7]. Since HPTLC is a faster and cheaper method, it could be used to quantify the level of caulerpin found in fish tissues or other organisms. No HPTLC method for the determination of caulerpin level in *Caulerpa* extracts has so far been reported.

The validation parameters were also tested in this study. In the literature, the LOD ranges of the secondary metabolites found in extracts detected by HPTLC were found between 5.25 ng spot^−1^ and 29.30 µg [33,34,35,36,37,38,39,40,41]. Furthermore, the LOQ ranges of the secondary metabolites found in extracts detected by HPTLC were found between 15.29 ng spot^−1^ and 97.83 µg [33,34,35,36,37,38,39,40,41]. Additionally, in the study by Turhan and Cavas (2019), the LOD and LOQ values of caulerpin were determined using HPLC and they found the LOD, LOQ, recovery, and precision values for 235 nm as 0.00179 g/L, 0.00543 g/L, 108.8%, and 7.8%, respectively [21]. In our study, the recovery was found as 89.81%. According to the ICH guidelines, the recovery (%) range of the analyte should be 70–130 (%). Furthermore, according to European Commission Health and Consumer Protection (ECHA) guidelines, the suggested precision value (R.S.D. %) is less than 20%. In our study, we found the R.S.D. at 19.2%.

The MALDI-TOF/MS result indicated that the fragment found in all samples at 398 m/z is caulerpin. Furthermore, the fragment found in the spectrum above 871 m/z could be chlorophyll α allomers (chlorophyll a without the magnesium ion, HO-chlorophyll a without the magnesium ion, HO-lactonechlorophyll a without the magnesium ion). In the literature, Yap et al. studied the antioxidant and antibacterial properties of *C. racemosa* and *C. lentillifera* extracts and identified the compounds found in these species [42]. They mentioned the properties of alloxhantin, which is a carotenoid, such as anti-inflammatory properties. Additionally, in the same study, they mentioned the biological properties of pheophorbide-a, such as inducing apoptosis in human hepatocellular carcinoma cells and antioxidant and anti-inflammatory activities [42]. Similar fragments were observed in our study.

Colorectal cancer (CRC) is the third most common cancer type and the second cause of cancer-related death worldwide and investigating, and using novel marine-sourced natural compounds as CRC treatment agents is crucial. Medicinal plants are defined as a “treasure trove” of biologically active molecules (chemicals) against various human diseases [43]. Caulerpin was previously used against different cancer cell lines and also colorectal cancer lines [10,44,45,46,47,48]. HT-29 and HCT-116 are the most commonly used for in vitro colorectal cancer research [26]. In this current study, because of this reason and their different genetical background (HT-29 cell line is *BRAF*-mutated colorectal adenocarcinoma; on the other hand, the HCT-116 cell line is *KRAS*-mutated colon carcinoma), we have selected these lines. Due to the molecular differences in cell lines, and also related culture conditions, various drug (caulerpin) responses can be observed. For instance, in this study, the IC_50_ values of these two cell lines were found to be different (HT-29 IC_50_: 179 µM and HCT-116 IC_50_: 119 µM). In the study of Yu et al. (2016), IC_50_ values of caulerpin against CRC cells were found to range from 20 to 31 µM [48]. The main reason for the difference in the IC_50_ values between our results compared to the obtained results from the previous study carried out by using HCT-116 and HT-29 cell lines could be the different cell viability assay kit that they have used. In the study of Yu et al. (2016), the CCK8 Assay kit was used for the measurement of cell viability, which is a more sensitive reagent than WST-1. The other reason for the difference in the IC50 values could be the medium that they used. The intensity of cell viability reagents is affected by the culture medium [49]. In the study of Yu et al. (2016), DMEM was used as growth media; on the other hand, in our study, HCT-116 and HT-29 cells were grown in McCoy’s 5A media, as suggested by ATCC. On the other hand, their main aim was to investigate the metabolic reprogramming and AMPKα1 pathway activation activity of caulerpin. This study aimed to investigate the inhibitory activity of caulerpin against metastatic processes involving migration and invasion. The first step of metastasis is the migration of cancer cells. Exposure to caulerpin inhibited the HCT-116 and HT-29 cell migration in a dose-dependent manner. Furthermore, caulerpin inhibited the HCT-116 and HT-29 cell invasion through Matrigel for 48-h.

Invasive species generally affect ecosystems and infrastructures detrimentally. Thus, utilizing the invasive species with their biotechnological and/or clinical properties is crucial. *Caulerpa cylindracea* Sonder, an invasive marine macroalga, has exhibited an impressive and continuous expansion since its first observation in Tunisia. Rather than eradication, investigating the “treasure trove” of invasive *C. cylindracea* and utilizing the chemicals as marine drugs is strongly recommended.

In terms of study limitations, the time of the maceration was quite short. The results of the maceration yield were calculated; on the other hand, it should be changed, and the effect of the time on the extraction yield should be evaluated. Additionally, although the caulerpin level found in invasive CCM was found as the highest value, from a future perspective, different *Caulerpa* samples might be studied. The difference between the extraction methods that were applied should be repeated. Future studies on the novel extraction and modifications in HPTLC methods will need further exploration.

## 4. Materials and Methods

### 4.1. Instrumentation and Chemicals

HPTLC analysis was studied using Camag Linomat V and Camag TLC-Scanner 3. TLC was performed on 20 × 10-cm plates (Merck, silica gel60 F254 TLC plates). Caulerpin samples were prepared fresh in diethyl ether and were applied to TLC by a semi-automatic sample applicator (Camag Linomat V, Muttenz, Switzerland). The twin horizontal chamber (Camag) was used for development. A Camag TLC-Scanner 3 densitometer was used to quantify the bands on the plates using a tungsten source (absorbance reflection mode). The slit dimensions were set as 6 mm and 0.3 mm in length and width, respectively. The scanning rate was set as 20-mm s^−1^. Diethyl ether and petroleum ether (Merck, analytical grade) were used for the extraction of the samples and used as the mobile phase. Ethyl acetate (high purity, Tekkim) was used for the maceration of samples.

### 4.2. Caulerpa Sampling

Specimens of green invasive seaweed *C. cylindracea* (Sonder) were collected from Dikili (İzmir-Türkiye) (39°12′19.31″N, 26°85′28.08″E) at a depth of 0–1 m. The samples were transferred into seawater through the laboratory and washed with water to separate them from epiphytes, sediments, and associated organisms such as seaweeds, seagrasses, and mollusks. Then, the invasive *C. cylindracea* samples were dried at room temperature. Dried and powdered *C. lentillifera* (from Shaanxi, China) samples were purchased from China (Xi’an Aladdin Biological Technology Co., Ltd., Xi’an, China).

### 4.3. Extraction of Caulerpa *spp.* and Isolation of Caulerpin

In this study, six different *Caulerpa* spp. extracts were prepared with the method of Aguilar-Santos (1970), with some modifications. Dried and powdered *C. cylindracea* samples (55 g) were extracted with diethyl ether (350 mL) using a Soxhlet apparatus for 8 h and re-extracted without removing the extract and residue with diethyl ether using a Soxhlet apparatus for 8 h more. Then, the extract was filtered and evaporated (hereafter, this extract is called CC48). The residues of *C. cylindracea* samples were re-extracted with diethyl ether using a Soxhlet apparatus for 8 h without removing the extract and residue. After 16 h, the extract was filtered and concentrated (hereafter, this extract is called CC72; yield 0.145%). Dried and powdered *C. lentillifera* samples (124 g) were extracted with diethyl ether (350 mL) using a Soxhlet apparatus for 8 h and re-extracted without removing the extract and residue with diethyl ether using a Soxhlet apparatus for 8 h more. Then, the extract of the non-invasive sample was filtered and evaporated (hereafter, this extract is called CL48). The residues of *C. lentillifera* samples were re-extracted with diethyl ether using a Soxhlet apparatus for 8 h without removing the extract and residue. After 16 h, the extract was filtered and concentrated (hereafter, this extract is called CL72). Then, *C. cylindracea* samples (1 g) were macerated with ethyl acetate (10 mL) and concentrated to 200 uL (hereafter, this extract is called CCM). *C. lentillifera* samples (1 g) were macerated with ethyl acetate (10 mL) and concentrated (hereafter, this extract is called CLM). Additionally, CC48 was set aside to cool for 48 h at −18 °C and red crystals of caulerpin were obtained.

### 4.4. Optimization of HPTLC Analysis

Caulerpin samples were applied to plates (1 µL per band), and band length and widths were set as 6 mm and 0.3 mm, respectively. The distance between bands, distance from the plate side, and distance from the bottom of the plate were 8.2 mm, 10 mm, and 15 mm, respectively. The plates (20 × 10 cm) were developed in a horizontal chamber. Petroleum ether-diethyl ether (1:1, *v/v*) was used as the mobile phase [21]. Then, 10 mL of mobile phase was added to the chamber to equilibrate the system 20 min before inserting the TLC plate, and after each analysis, the mobile phase was removed. The plates were scanned at 330 nm. The slit dimension was set as 6 mm × 0.3 mm. The scanning rate was 20 mm s^−1^. After the development of the caulerpin and extracts, the bands were identified by their retention factor (Rf) values.

### 4.5. Method Validation of Caulerpin

The method validation of caulerpin was performed using the International Council for Harmonization (ICH, 2005) Q2 (R1) recommendations. The calibration curve was plotted for 25, 50, 75, 100, and 500 ng µL^−1^ (the concentration in the bands) caulerpin after the optimization of the HPTLC procedure. The concentration of caulerpin versus the peak area was used to construct the calibration curve. The blank method was used to calculate the limit of detection (LOD) and limit of quantitation (LOQ) values for caulerpin [50]. In this method, multiplying by three standard deviations of the response of the ten different analyses of ether (blank sample) divided by the slope was defined as the LOD of the Caulerpin. LOQ was calculated as 3.3-fold of the LOD values. To calculate the accuracy, freshly prepared caulerpin standards with the concentrations of 25, 50, and 100 ng µL^−1^ were analyzed three times (three concentrations, three replicates) and were estimated as percentage recovery (R%). Precision was calculated by the maximum concentration level (500 ng µL^−1^) of caulerpin with six replicates. It was represented as the relative standard deviation percentage (R.S.D. %) [51].

### 4.6. Cell Lines and Reagents

The cell lines HCT-116 (colorectal carcinoma), HT-29 (colorectal adenocarcinoma), NIH-3T3 (*Mus musculus* fibroblast), and HDF (primary dermal fibroblast; normal, human) were obtained from American Type Culture Collection (ATCC). McCoy’s 5A and fetal bovine serum (FBS) were purchased from Cegrogen, Ebsdorfergrund, Germany. WST-1, Cell Proliferation Reagent, was purchased from Roche, Indianapolis, IN, USA. Matrigel (growth factor reduced, basement membrane matrix) was purchased from Corning, Somerville, MA, USA. Dimethyl sulfoxide (for cell culture) was purchased from PanReac AppliChem, Chicago, IL, USA.

### 4.7. Cell Culture

The HCT-116 (ATCC, CCL-247) cell line was routinely maintained in McCoy’s 5A supplemented with 10% fetal bovine serum, 1% L-glutamine, and 1% penicillin/streptomycin; and the HT-29 (ATCC, HTB-38) cell line was routinely maintained in McCoy’s 5A supplemented with 10% fetal bovine serum and 1% penicillin/streptomycin. NIH-3T3 (ATCC, CRL-1658) and HDF (primary dermal fibroblast; normal, human ATCC, PCS-201-012) cell lines were maintained in DMEM/HamsF12 supplemented with 10% fetal bovine serum and 1% penicillin/streptomycin. Cell lines were incubated at 5% CO_2_ and 37 °C temperatures.

### 4.8. Cytotoxicity Assay

The cells (4 × 10^4^ each) were seeded in a 96-well plate, and after 24-h incubation, cells were treated with different concentrations of caulerpin for up to 48 hr. After 24- and 48-h incubation, WST-1 was used and incubated for 2.5 h. Following incubation, absorbance was taken at 450/630 nm wavelength using a spectrophotometer (Variskan Lux, ThermoScientific, Waltham, MA, USA).

Inhibition concentration (IC) 50 and IC10 values were calculated by using the formulas of [(Tx − Tz)/Tz] × 100 = −50 and −10, respectively. (C is the control, Tx is a time of the absorbance measurement of caulerpin concentrations, i.e., 24 and 48 h, and Tz is the absorbance time zero) [49].

### 4.9. Colony Formation Assay

Cell lines seeded 5 × 10^2^ in a 6-well plate were used. After 48 h of seeding, IC_50_ and IC_10_ doses of caulerpin isolated from invasive *C. cylindracea* were applied. The medium in each well was reintroduced at the end of each 48 h after treatment with a caulerpin. After 14 days, cells were washed 2 times with cold methanol, treated with 4% PFA for 20 min, and stained with 0.5% in 2.5% methanol.

### 4.10. Migration Assay

HCT-116 and HT-29 cells (3 × 10^5^ cells/well) were seeded in a 6-well plate and incubated until 90% confluency. Before the cells were treated with caulerpin, the media was removed, and FBS-starvation media was used for 24 h. The wells were straightly scratched with a sterile pipette tip across the center of the well to simulate a wound and the IC_50_ and IC_10_ doses of caulerpin were added. After treatment, the images were taken at 0, 24, and 48 h by using an Axio inverted light microscope (Zeiss). The widths of the wounds from at least 2 different fields of the wells were measured using the ZEISS ZEN microscope software and the widths were quantified.

The wound width percentage was calculated by using the formula of [(Wound width Tz − Wound width Tx)/Wound width Tz] × 100 (Tz is time zero, Tx is the time of the measurement) [52].

### 4.11. Invasion Assay

The invasion experiments were performed using an xCELLigence RTCA DP Instrument with CIM-plate 16 (Roche Diagnostics GmbH, Mannheim, Germany). Before the experiment, the cells were suspended in an FBS-starvation medium for at least 6 h. The upper compartment was coated with 20 μL Matrigel (growth factor reduced, basement membrane matrix, BD Biosciences, Erembodegem, Belgium) at a 1:30 ratio 4 h before the addition of cell suspension for invasion assay. The lower compartment was supplemented with chemoattractant (20% FBS-containing media) to create a concentration gradient. HCT-116 and HT-29 cells (4 × 10^4^ cells/well) suspended in the FBS-starvation medium were added to the upper compartment. Caulerpin with different concentrations (IC_50_ and IC_10_ concentrations of caulerpin calculated by WST-1 assay analysis) was added to the upper compartment. The impedances were recorded every 15 min and monitored for 48 h xCELLigence RTCA software vs.1.2.1. The invasion of cells at the 6th, 24th, and 48th hours was calculated as a percentage of CI compared to the control group by using Graphpad Prism.

### 4.12. Apoptosis Analysis

For the apoptosis assay, the Hoechst (33342)/PI staining procedure was used. First, 10,000 cells were seeded in 96-well plates and incubated for 48 h with IC10 and IC50 doses. After harvesting the HCT-116 and HT-29 cells, cold PBS was used for washing (three times for 5 min), cells were fixed by using methanol and stained using Hoechst 33,342 (1 µg mL^−1^) and PI (5 µg mL^−1^). After incubation for 30 min at 37 °C in the dark, cells were washed with cold PBS (three times for 5 min). The images were taken with a confocal microscope (ZEISS, Germany) [53].

The apoptosis percentage (%) was calculated as the PI-positive stained cells found in total Hoechst-stained cells.

### 4.13. Statistical Analysis

Student’s *t*-test (*t*-test) was used for the comparison of means between the groups (extracts; heated or macerated at room temperature). Statistical difference was set at *p* = 0.05. The results of in vitro tests were analyzed either by two-way analysis of variance (two-way ANOVA), one-way analysis of variance (one-way ANOVA), or Student’s *t*-test as required by the experimental system, and differences were considered significant at *p* < 0.05 by using GraphPad Prism.

## 5. Conclusions

The *Caulerpa* genus is an important macroalgae genus with 96 different species, including invasive members. The secondary metabolites of invasive *Caulerpa cylindracea* attract attention due to their bioactivities. Overall, in this study, HPTLC was evaluated as a quick and effective method to determine caulerpin levels in *Caulerpa* tissues. The data obtained from HPTLC analysis were effectively built for extracts of both non-invasive and invasive *Caulerpa* spp. HPTLC demonstrated a new approach to investigating the similarity between the species and extraction methods based on their features in HPTLC chromatograms. Moreover, qualitative, and quantitative analyses were performed to identify the ingredients of the extracts. Changes in the extraction method of *Caulerpa* samples could influence the HPTLC analysis. Proliferation–invasion–migration–apoptosis and colony-formation-based anticancer properties of caulerpin isolated from *C. cylindracea* against CRC are shown via HCT-116 and HT-29 cell lines, and results were compared by NIH-3T3 and HDF cell lines. Invasive species in the Mediterranean Sea may be utilized to discover new bioactive compounds that will be evaluated in pharmacy and medicine. This paper proposes an alternative utilization method for invasive *C. cylindracea* due to significant caulerpin content compared to non-invasive *C. lentillifera.* Invasive species may differ in their secondary metabolites in terms of concentration and modification during the adaptation period within their invaded habitats, which will need the development of new analytical methods such as those mentioned in this paper. In this paper, caulerpin found in invasive *C. cylindracea* and non-invasive *C. lentillifera* extracts were analyzed and quantified using HPTLC for the first time. Compared to other chromatographic (especially HPLC-based) methods, HPTLC is a quick method for quantification of caulerpin and consumes less solvent. Among the two different extraction procedures (soxhlet and maceration) prepared from two different species (invasive and non-invasive), the caulerpin levels were found higher in the invasive species. Since there was a solubility problem in the CL48 samples, the caulerpin level could not be measured and the caulerpin level was found below LOD. The extraction method of CL samples could be revised for further studies. The anticancer activity of caulerpin against colorectal cancer cell lines was also tested. To elucidate the mechanistic insight of caulerpin, additional apoptotic biomarkers should be tested for further studies. In conclusion, caulerpin was found to be an effective and promising marine compound from invasive species against CRC. On the other hand, further studies for translational approaches are recommended.

## Figures and Tables

**Figure 1 marinedrugs-20-00757-f001:**
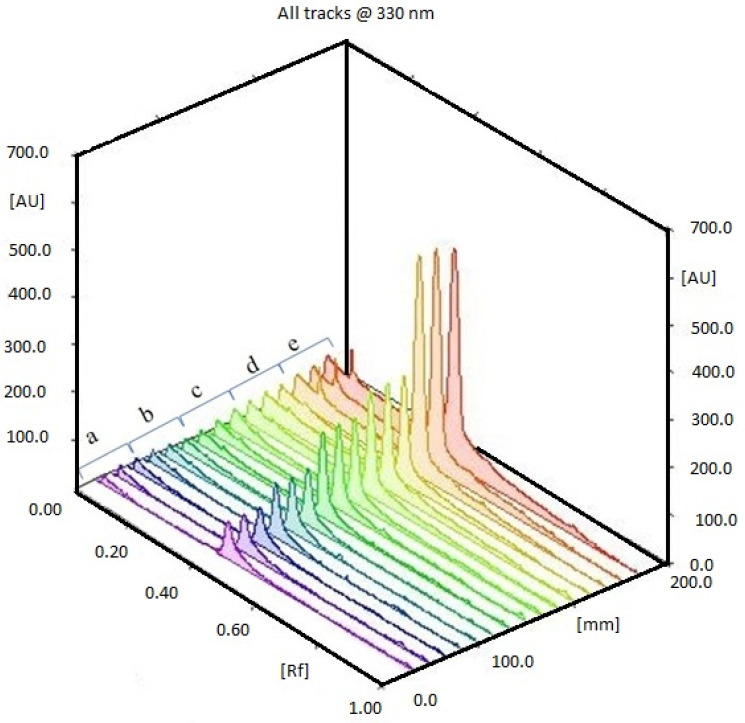
HPTLC chromatogram of (a) 25 ng µL^−1^, (b) 50 ng µL^−1^, (c) 75 ng µL^−1^, (d) 100 ng µL^−1^, (e) 500 ng µL^−1^ caulerpin.

**Figure 2 marinedrugs-20-00757-f002:**
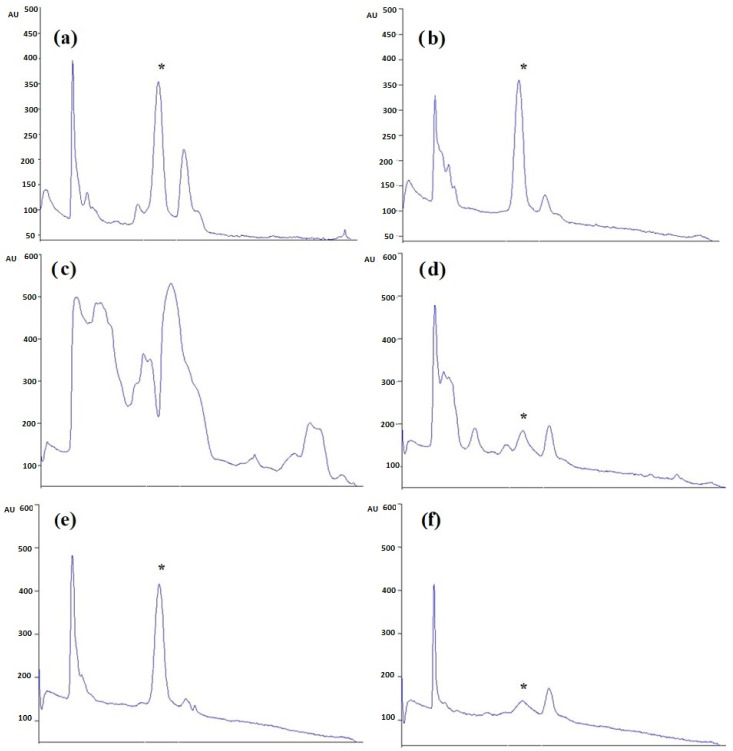
HPTLC chromatogram of *Caulerpa* extracts (**a**) CC48, (**b**) CC72, (**c**) CL48, (**d**) CL72, (**e**) CCM, and (**f**) CLM at 330 nm. * indicates the caulerpin (Rf: 0.41).

**Figure 3 marinedrugs-20-00757-f003:**
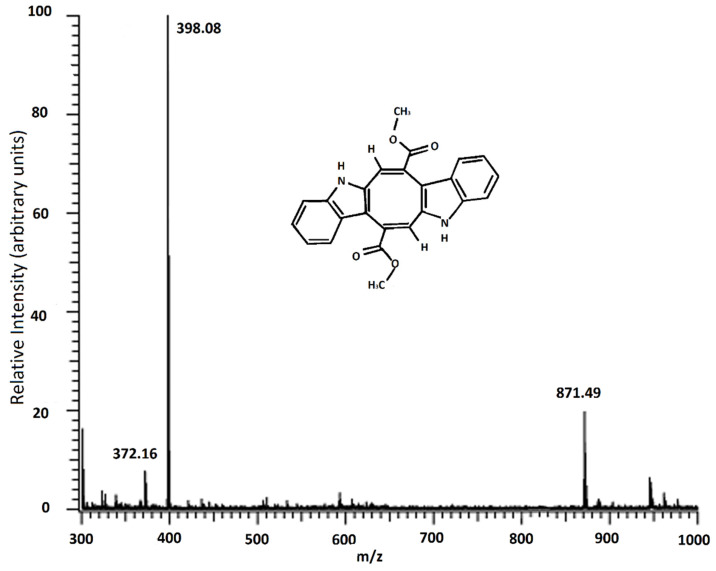
MALDI-TOF/MS result of caulerpin.

**Figure 4 marinedrugs-20-00757-f004:**
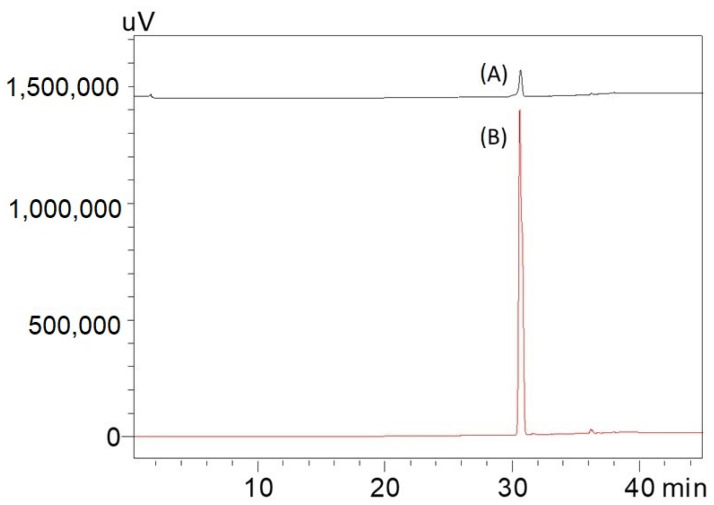
Comparing HPLC chromatogram of 10 and 100 pm caulerpin. Conditions: Inertsil C18 column (150 × 10 mm, 5 μm), mobile phase: MeOH: water (non-linear gradient; Turhan and Cavas, 2019) wavelength 235 nm, flow rate: 1 mL/min, column temperature: 25 °C injection volume: 50 μL, loop volume: 100 μL. Caulerpin was dissolved in methanol. (A) 10 ppm caulerpin filtered by using 0.45 μm membrane filter before injection; (B) 100 ppm caulerpin filtered by using 0.45 μm membrane filter before injection.

**Figure 5 marinedrugs-20-00757-f005:**
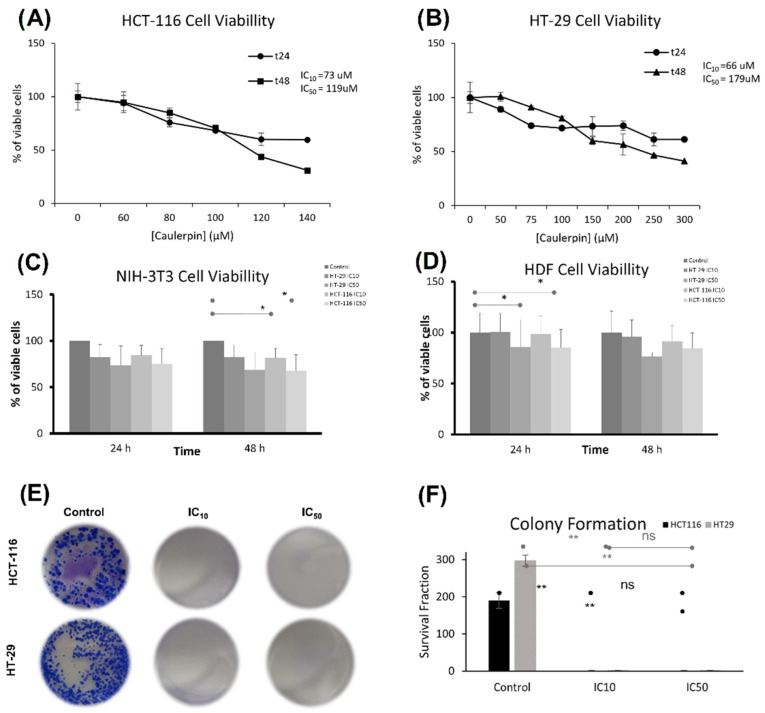
Caulerpin induced cytotoxicity in colorectal cell lines. (**A**) HCT-116 and (**B**) HT-29 cell lines were treated with different caulerpin concentrations for 24 and 48 h. (**C**) NIH-3T3 and (**D**) HDF cell lines were treated with IC_10_ and IC_50_ doses of caulerpin for 24 and 48 h. (**E**,**F**) Colony formation assay was performed in HCT-116 and HT-29 cell lines exposed to caulerpin at IC_10_ and IC_50_ concentrations. Data represent the mean ± SEM of three independent experiments. Statistical differences were analyzed with the student’s *t*-test (* indicates *p* < 0.05, ** indicates *p* < 0.001, ns indicates not significant).

**Figure 6 marinedrugs-20-00757-f006:**
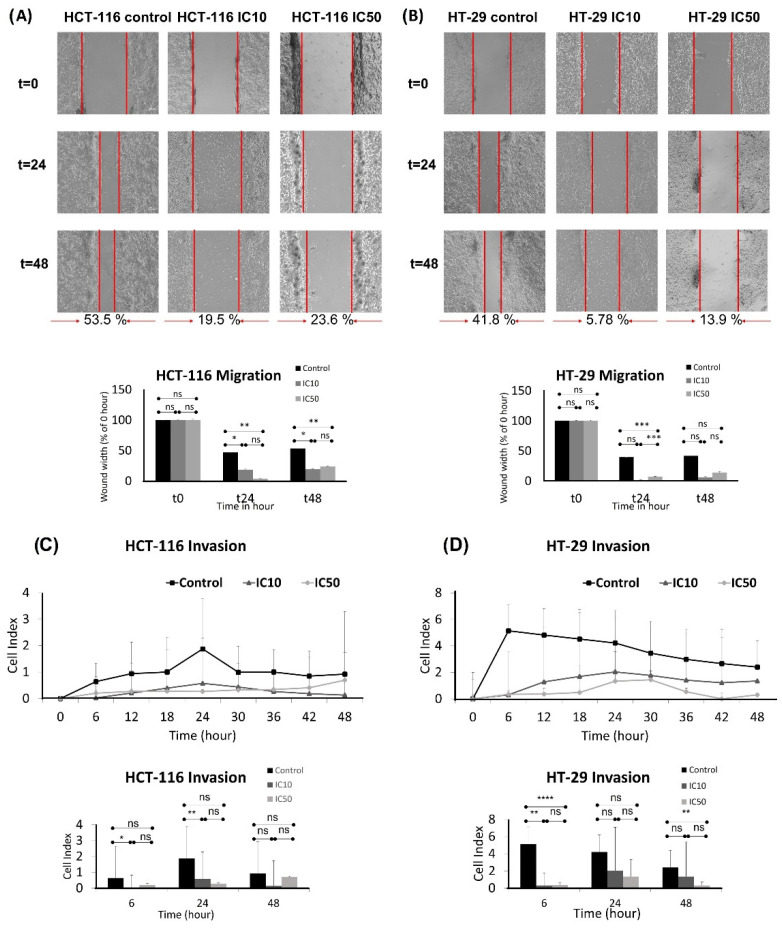
Caulerpin inhibited the migrative and invasive properties of colorectal cell lines. Inhibition of migration of HCT-116 (**A**) and HT-29 (**B**) with the treatment of IC_10_ and IC_50_ doses for 24 and 48-h. The wound widths were quantified in lower panels. Inhibition of invasion of HCT-116 (**C**) and HT-29 (**D**) cells through a Matrigel-coated CIM plate with the treatment of IC_10_ and IC_50_ doses of caulerpin for 48-h. The real-time cell index of invasion of HCT-116 and HT-29 cell lines was given in the upper panel; the end-point results were given in the lower panel. Data represent the mean ± SEM of three independent experiments. Statistical differences were analyzed with a two-way ANOVA test (* indicates *p* < 0.005, ** indicates *p* < 0.001, *** indicates *p* < 0.0001, **** indicates *p* < 0.00001, ns indicates not significant).

**Figure 7 marinedrugs-20-00757-f007:**
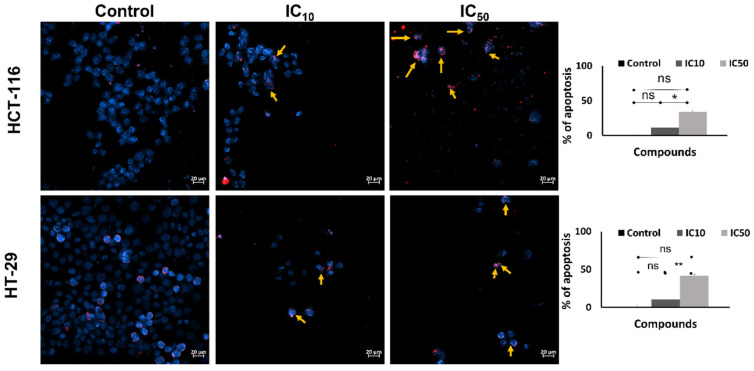
Induction of apoptosis in CRC by caulerpin. Hoechst/propidium iodide (PI) staining of HCT-116 and HT-29 cell lines (scale bar: 20 µm) treated with different concentrations (IC_10_ and IC_50_ doses) of caulerpin for 48-h. The Hoechst and PI merged images are given in the left panel and the apoptotic cell percent are given in the right panel. Data represent the mean ± SEM of three independent experiments. Statistical differences were analyzed with the student’s *t*-test (* indicates *p* < 0.05, ** indicates *p* < 0.001, ns indicates not significant).

**Table 1 marinedrugs-20-00757-t001:** Caulerpin levels found in *Caulerpa* extracts.

	Concentration (ng µL^−1^)	Concentration (µg g^−1^)
CC48	453.46 ± 21.11	108.83 ± 5.07
CC72	402.04 ± 18.93	96.49 ± 4.54
CL48	n.d.	n.d.
CL72	35.16 ± 16.08 (below the LOQ)	8.44 ± 3.86 (below the LOQ)
CCM	562.14 ± 26.43	112.43 ± 5.29
CLM	42.09 ± 8.21 (below the LOQ)	8.42 ± 1.64 (below the LOQ)

**Table 2 marinedrugs-20-00757-t002:** HPTLC validation parameters of caulerpin.

Validation Parameters	Caulerpin (330 nm)
Calibration equation	y = 28.872x + 3233.4
Linearity (R^2^)	0.9635
Slope	28.872
Shift	3233.4
Range (ng band^−1^)	25–500
LOD (ng band^−1^)	20.47
LOQ (ng band^−1^)	67.56
Recovery (%)	89.81
RSD (%)	19.2

## Data Availability

The data presented in this study are available on request from the corresponding author. The data are not publicly available.

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
