# Peer review of "An Efficient and Quick Analytical Method for the Quantification of an Algal Alkaloid Caulerpin Showed In-Vitro Anticancer Activity against Colorectal Cancer"

_marinedrugs, 2022, doi:10.3390/md20120757_

Round 1
Reviewer 1 Report
Mert-Ozupek et al., study was designed to isolate and characterize caulerpin from C. cylindracea (invasive) and C. lentillifera (non-invasive) samples as well as evaluate its anticancer activity against colorectal cancer cells.
I think the data presented in this manuscript is interesting, however, this manuscript in its present form, it is not yet fully acceptable for publication. I recommend that the authors should address the below comments before this manuscript could be considered for publication.
Comments:
1. Authors need to rewrite the abstract section so that the most important results and conclusions are highlighted.
2. Extraction method details is lacking; volume of solvent used, how many grams of plant and yield of extracts.
3. To investigate the anti-cancer activity, the author should provide evidence for lack of activity against non-cancerous cells to compare the selectivity of the caulerpin. The positive control of cytotoxicity assay should be added.
4. In section 4.12. Apoptosis analysis, more details are needed such as Hoechst (33342)/PI concentrations, PFA abbreviation and staining period. (Hoechst (33342)/PI stained and fixed), I think the fixation step is before staining step. Therefore, the complete section needs to be rewritten.
5. Figure 6 does not justify apoptosis induction at all, can the authors show with arrows formation of apoptotic bodies and or nuclear condensation and disintegration as expected in Hoechst stained nuclei.
6. Authors need to include quantitative assays for apoptosis induction measurements using flow cytometry analysis such as Annexin-FITC/PI assay. The authors can also elucidate the anticancer action of caulerpin with additional apoptotic markers such as Bcl-2, Bax as well as caspases that may provide a mechanistic insight.
7. Authors must clearly explain how the IC50 and IC10 was obtained, explain how they quantified the apoptosis percentage and how they calculated the wound widths.
8. In the discussion part, authors need to justify why they selected these cells (colorectal cells), the high difference in the IC50 values (between HCT-116 and HT- 29 cell lines) in this study as well as to compare the obtained results with previous studies.
9. The Conclusion part is poorly written. Authors should indicate what is new and innovative in the investigation, gaps in this investigation and how to close them as well as indicating the holistic conclusion and recommendations.
10. The manuscript requires language editing as numerous grammatical errors present.
Author Response
The comments of the reviewer#1:
I think the data presented in this manuscript is interesting, however, this manuscript in its present form, it is not yet fully acceptable for publication. I recommend that the authors should address the below comments before this manuscript could be considered for publication.
Authors need to rewrite the abstract section so that the most important results and conclusions are highlighted.
Our response:
First of all, we would like to thank reviewer#1 for her/his valuable comments. We have revised the abstract section of the submitted version of the manuscript. The revised abstract was given below:
Biological invasion is the successful spread and establishment of a species in a novel environment which adversely affects the biodiversity, ecology, and economy. Both invasive and non-invasive species of the Caulerpa genus secrete more than thirty different secondary metabolites. Caulerpin is one of the most common secondary metabolites in Genus Caulerpa. In this study, caulerpin found in invasive C. cylindracea and non-invasive C. lentillifera extracts were analyzed, quantified, and compared using High-Performance Thin Layer Chromatography (HPTLC) for the first time. The anticancer activities of caulerpin against HCT-116 and HT-29 colorectal cancer (CRC) cell lines were also tested. Caulerpin levels were found higher in invasive form (108.83 ± 5.07 µg mL-1 and 96.49 ± 4.54 µg mL-1). Also, caulerpin isolated from invasive Caulerpa decreased cell viability in a concentration-dependent manner (IC50 values were found between 119 and 179 µM) inhibits invasion-migration, and induces apoptosis in CRC cells, in comparison, no cytotoxic effects on the normal cell lines (HDF and NIH-3T3). In conclusion, HPTLC is a quick and novel method to investigate the caulerpin levels found in Caulerpa extracts, and this paper proposes an alternative utilization method for invasive C. cylindracea due to significant caulerpin content compared to non-invasive C. lentillifera.
The comments of the reviewer#1:
Extraction method details is lacking; volume of solvent used, how many grams of plant and yield of extracts.
Our response:
We have revised the related section as given below:
“In this study, six different Caulerpa spp. extracts were prepared and the method of Aguilar-Santos (1970) with some modifications. Dried and powdered C. cylindracea samples (55 g) were extracted with diethyl ether (350 mL) using a Soxhlet apparatus for 8 h and re-extracted without removing the extract and residue with diethyl ether using a Soxhlet apparatus for 8 h more. Then, the extract was filtered and evaporated (hereafter, this extract is called CC48). The residues of C. cylindracea samples were re-extracted diethyl ether using a Soxhlet apparatus for 8 h without removing the extract and residue. After 16 h, the extract was filtered and concentrated (hereafter, this extract is called CC72; yield 0.145%). Dried and powdered C. lentillifera samples (124 g) were extracted with diethyl ether (350 mL) using a Soxhlet apparatus for 8 h and re-extracted without removing the extract and residue with diethyl ether using a Soxhlet apparatus for 8 h more. Then, the extract of non-invasive sample was filtered and evaporated (hereafter, this extract is called CL48). The residues of C. lentillifera samples were re-extracted with diethyl ether using a Soxhlet apparatus for 8 h without removing the extract and residue. After 16 h, the extract was filtered and concentrated (hereafter, this extract is called CL72). Also, C. cylindracea samples (1 g) were macerated with ethyl acetate (10 mL) and concentrated to 200 uL (hereafter, this extract is called CCM). C. lentillifera samples (1 g) were macerated with ethyl acetate (10 mL) and concentrated (hereafter, this extract is called CLM). Also, CC48 was set aside to cool for 48 h at -18°C and red crystals of caulerpin were obtained.”
The comments of the reviewer#1:
To investigate the anti-cancer activity, the author should provide evidence for lack of activity against non-cancerous cells to compare the selectivity of the caulerpin. The positive control of cytotoxicity assay should be added.
Our response:
We would like to thank reviewer#1 for her/his valuable comments. We have re-designed our cell viability assay based on the study of Ru et al. (2022) with some modifications and non-cancerous cell lines (NIH-3T3, Mus musculus fibroblast cell line; HDF, Human dermal fibroblast cell line) were treated with caulerpin with dose- and time-dependent manner to investigate the activity against cell viability. The results were given below. On the other hand, the positive control could not be tested because of budget limitations. However, our results may lead us one step closer to elucidating the mechanism of action of caulerpin, thus positive control will be used in our further studies.
“For the NIH/3T3 cell line, the IC10 doses of the caulerpin did not significantly reduce the cell viability at 24th hour compared to the control group. However, for the NIH/3T3 cell line, the IC10 dose of the HCT-116 cell line (*p=0.04), and the IC50 dose (*p=0.04) dose of the HCT-116 cell line were significantly reduced viability at 48 hours compared to the control. On the other hand, the IC50 doses did not reach the half maximal viability at 48th hour (HCT-116 IC50 dose cell viability: 67.5% and HT-29 IC50 dose cell viability: 68.8%) (Figure 5C).
For the HDF cell line, the IC50 dose of the HT-29 cell line (*p=0.04), and the IC50 dose of the HCT-116 cell line (*p=0.04) significantly reduced the cell viability at 24th compared to the control group. Furthermore, the IC10 doses of the caulerpin did not significantly reduce the cell viability at 24th hour compared to the control group. On the other hand, the IC50 doses did not reach the half maximal viability at 48th hour (HCT-116 IC50 dose cell viability: 84.7% and HT-29 IC50 dose cell viability: 76.6%) (Figure 5D).”
- Ru, R., Guo, Y., Mao, J., Yu, Z., Huang, W., Cao, X., ... & Yuan, L. (2022). Cancer Cell Inhibiting Sea Cucumber (Holothuria leucospilota) Protein as a Novel Anti-Cancer Drug. Nutrients, 14(4), 786.
The comments of the reviewer#1:
In section 4.12. Apoptosis analysis, more details are needed such as Hoechst (33342)/PI concentrations, PFA abbreviation and staining period. (Hoechst (33342)/PI stained and fixed), I think the fixation step is before staining step. Therefore, the complete section needs to be rewritten.
Our response:
We have revised above-mentioned section as given below:
“For the apoptosis assay, the Hoechst (33342)/PI staining procedure was used. 10000 cells were seeded in 96-well plates and incubated for 48 h with IC10 and IC50 doses. After harvesting the HCT-116 and HT-29 cells, cold PBS was used for washing (3 times for 5 min), cells were fixed by using methanol and stained using Hoechst 33342 (1 µg mL-1) and PI (5 µg mL-1). After incubation for 30 min at 37°C in dark, cells were washed with cold PBS (three times for 5 min). The images were taken with a confocal microscope (ZEISS, Germany) (Wang et al., 2015).”
- Wang, X., Tan, T., Mao, Z. G., Lei, N., Wang, Z. M., Hu, B., ... & Wang, H. J. (2015). The marine metabolite SZ-685C induces apoptosis in primary human nonfunctioning pituitary adenoma cells by inhibition of the Akt pathway in vitro. Marine drugs, 13(3), 1569-1580.
The comments of the reviewer#1:
Figure 6 does not justify apoptosis induction at all, can the authors show with arrows formation of apoptotic bodies and or nuclear condensation and disintegration as expected in Hoechst stained nuclei.
Our response:
We have revised above mentioned figure as given below:
The comments of the reviewer#1:
Authors need to include quantitative assays for apoptosis induction measurements using flow cytometry analysis such as Annexin-FITC/PI assay. The authors can also elucidate the anticancer action of caulerpin with additional apoptotic markers such as Bcl-2, Bax as well as caspases that may provide a mechanistic insight.
Our response:
We would like to thank reviewer#1 for her/his valuable comments. For quantitative measurement of apoptosis, three visual fields for each treatment, and percentage (%) was calculated as the PI-positive stained cells found in total Hoechst-stained cells (Oh et al., 2019). Unfortunately, we could not perform above mentioned suggestions because of the budget limitation. However, we are planning to elucidate the anticancer action of caulerpin with additional apoptotic markers in our further projects.
The comments of the reviewer#1:
Authors must clearly explain how the IC50 and IC10 was obtained, explain how they quantified the apoptosis percentage and how they calculated the wound widths.
Our response:
We have revised the relevant part in the materials and method section:
“Inhibition concentration (IC) 50 and IC10 values were calculated by using the formulas of [(Tx-Tz) / Tz] x 100 = -50 and -10, respectively. (C is the control, Tx is a time of the absorbance measurement of caulerpin concentrations i.e. 24 and 48 h, and Tz is the absorbance time zero) (Präbst, et al., 2017).”
“The apoptosis was quantitated by three visual fields for each treatment, and percentage (%) was calculated as the PI-positive stained cells found in total Hoechst-stained cells (Oh et al., 2019).”
“The wound width percentage was calculated by using the formula of [(Wound width Tz- Wound width Tx) / Wound width Tz]×100 (Tz is time zero, Tx is the time of the measurement) (Buachan et al., 2014)”
- Präbst, K., Engelhardt, H., Ringgeler, S., & Hübner, H. (2017). Basic colorimetric proliferation assays: MTT, WST, and resazurin. In Cell viability assays (pp. 1-17). Humana Press, New York, NY.
- Oh, J. M., Kim, E., & Chun, S. (2019). Ginsenoside compound K induces ros-mediated apoptosis and autophagic inhibition in human neuroblastoma cells in vitro and in vivo. International journal of molecular sciences, 20(17), 4279.
- Buachan, P., Chularojmontri, L., & Wattanapitayakul, S. K. (2014). Selected activities of Citrus maxima Merr. fruits on human endothelial cells: enhancing cell migration and delaying cellular aging. Nutrients, 6(4), 1618-1634.
The comments of the reviewer#1:
In the discussion part, authors need to justify why they selected these cells (colorectal cells), the high difference in the IC50 values (between HCT-116 and HT- 29 cell lines) in this study as well as to compare the obtained results with previous studies.
Our response:
We have revised the discussion part as given below:
“HT-29 and HCT-116 are the most commonly used for the in-vitro colorectal cancer research [27]. In this current study, because of this reason and their different genetical background (HT-29 cell line is BRAF-mutated colorectal adenocarcinoma, on the other hand, the HCT-116 cell line is KRAS-mutated colon carcinoma), we have selected these lines. Due to the molecular differences in cell lines, and also related culture conditions can cause various drug (caulerpin) responses. For instance, in this study, the IC50 values of these two cell lines were found different (HT-29 IC50: 179 µM and HCT-116 IC50: 119 µM). In the study of Yu et al. (2016), IC50 values of caulerpin against CRC cells were found to range from 20 to 31 µM [49]. The main reason for the difference in the IC50 values between our results compared to the obtained results from the previous study carried out by using HCT-116 and HT-29 cell lines could be the cell viability assay kit difference that they used. In the study of Yu et al. (2016), the CCK8 Assay kit was used for the measurement of cell viability which is a more sensitive reagent than WST-1. The other reason for the difference in the IC50 values could be the medium that they used. The intensity of cell viability reagents is affected by the culture medium (Präbst et al., 2017). In the study of Yu et al (2016), DMEM was used as growth media, on the other hand, in our study, HCT-116 and HT-29 cells were grown in McCoy's 5A media as suggested by ATCC.”
Präbst, K., Engelhardt, H., Ringgeler, S., & Hübner, H. (2017). Basic Colorimetric Proliferation Assays: MTT, WST, and Resazurin. Cell Viability Assays, 1–17. doi:10.1007/978-1-4939-6960-9_1
The comments of the reviewer#1:
The Conclusion part is poorly written. Authors should indicate what is new and innovative in the investigation, gaps in this investigation and how to close them as well as indicating the holistic conclusion and recommendations
Our response:
We have revised the conclusion part and the paragraph appended to the end of the conclusion was given below:
“In this paper, caulerpin found in invasive C. cylindracea and non-invasive C. lentillifera extracts were analyzed and quantified using HPTLC for the first time. Compared to other chromatographic (especially HPLC-based) methods, HPTLC is a quick method for quantification of caulerpin and consumes less solvent. Among the two different extraction procedures (soxhlet and maceration) prepared from two different species (invasive and non-invasive), the caulerpin levels were found higher in invasive species. Since there is a solubility problem in CL48 samples, the caulerpin level could not be measured and the caulerpin level was found below LOD. The extraction method of CL samples could be revised for further studies. The anticancer activity of caulerpin against colorectal cancer cell lines was also tested. To elucidate the mechanistic insight of caulerpin, additional apoptotic biomarkers should be tested for further studies. In conclusion, caulerpin was found as an effective and promising marine compound from invasive species against CRC. On the other hand, further studies for translational approaches are recommended.”
The comments of the reviewer#1
The manuscript requires language editing as numerous grammatical errors present.
Our response:
Grammatical errors and typos were edited
Author Response
The comments of the reviewer#2:
Herein, the authors aimed to determine of caulerpin level in different extracts of invasive C. cylindracea and non-invasive C. lentillifera.by using HPTLC quantitatively, validation of the HPTLC method by using International Council for Harmonization (ICH) Q2 94 (R1) recommendations, the characterization of isolated samples by using MALDI-TOF/MS 95 and investigate the biological activities of caulerpin against CRC cell lines using in vitro 96 methods. Many points should be discussed:
Title: It should be more impressive, shorter reflecting the importance of the study.
Our response:
First of all, we would like to thank reviewer#2 for her/his valuable comments. We have revised the title to “A novel and quick analytical method of anticancer associated caulerpin from invasive and non-invasive Caulerpa spp.”
The comments of the reviewer#2:
Introduction: What is the novelty of the research and how it is beneficial to the researchers? This point should be comprehensively clarified in the last paragraph in the introduction
Our response:
We have revised the last paragraph in the introduction part and the revised form was given below:
“Compared to the other chromatographic methods (especially HPLC), HPTLC is a novel and especially quick method to determine the caulerpin level found in invasive Caulerpa cylindracea extracts quantitatively and also caulerpin is a promising marine drug for CRC treatment.”
The comments of the reviewer#2:
Introduction: The introduction is too long and carries unnecessary information, it should be summarized to the point
Our response:
We have revised the introduction and removed the unnecessary information as reviewer#2 suggested.
The comments of the reviewer#2:
Materials and methods: It is wee-written
Our response:
The materials and methods section was revised, and new references were added.
The comments of the reviewer#2:
Results: The structure of the discussed compound (Caulerpin) should be drawn using chemdraw or any software
Our response:
The chemical structure of the caulerpin was drawn and added to MALDI-TOF/MS result.
The comments of the reviewer#2:
Figure 3 (HPTLC chromatogram of Caulerpa extracts) all the chromatograms don’t show the time o the X-axis in addition Figure 3 C is so bad and unresolved and does not show the peak for Caulerpin, it should be re-measured as this is totally unacceptable
Our response:
The reviewer is right. In Figure 3C, there is a solubility problem. On the other hand, we could not measure any caulerpin in the CL48 extract, furthermore, in CL48, the caulerpin level is found below LOD. The extraction method of CL samples could be revised for further studies
.
The comments of the reviewer#2:
Discussion : It should not contain figures and tables, table 3 and figure 7 must be included in the result section otherwise they are unimportant t be added as such
Our response:
Figure 7 is included in the results (MALDI-TOF/MS section). Also, Table 3 is removed from the manuscript.
The comments of the reviewer#2:
Additional points: Slight spelling and grammatical errors so should be checked by a native speaker
Our response:
Grammatical errors and typos were edited.
The comments of the reviewer#2:
Additional points:References should be accurately adjusted and avoid the presence of capital letters within the references
Our response:
The references were revised.
Round 2
Reviewer 1 Report
The authors satisfactory answered to the reviewer's suggestions. The manuscript is now suitable for publication in Marine Drugs.
Author Response
Dear Reviewer,
Thank you.
Reviewer 2 Report
No additional comments
Author Response
Dear Reviewer,
Thank you.